# Whole-genome sequencing of acral melanoma reveals genomic complexity and diversity

Felicity Newell [1✉], James S. Wilmott[2], Peter A. Johansson[1], Katia Nones[1], Venkateswar Addala [1,3], Pamela Mukhopadhyay[1], Natasa Broit [1,3], Carol M. Amato [4], Robert Van Gulick[4], Stephen H. Kazakoff [1], Ann-Marie Patch [1], Lambros T. Koufariotis[1], Vanessa Lakis[1], Conrad Leonard [1], Scott Wood [1], Oliver Holmes[1], Qinying Xu[1], Karl Lewis[4], Theresa Medina[4], Rene Gonzalez[4], Robyn P. M. Saw [2,5,6], Andrew J. Spillane [2,5,7], Jonathan R. Stretch[2,5,6], Robert V. Rawson[2,5,6,8], Peter M. Ferguson[2,5,6,8], Tristan J. Dodds[2], John F. Thompson [2,5,6], Georgina V. Long [2,5,7], Mitchell P. Levesque[9], William A. Robinson[4], John V. Pearson[1], Graham J. Mann[2,10,11], Richard A. Scolyer [2,5,6,8], Nicola Waddell [1,3,12] & Nicholas K. Hayward [1,12]

To increase understanding of the genomic landscape of acral melanoma, a rare form of melanoma occurring on palms, soles or nail beds, whole genome sequencing of 87 tumors with matching transcriptome sequencing for 63 tumors was performed. Here we report that mutational signature analysis reveals a subset of tumors, mostly subungual, with an ultraviolet radiation signature. Significantly mutated genes are *BRAF*, *NRAS*, *NF1*, *NOTCH2*, *PTEN* and *TYRP1*. Mutations and amplification of *KIT* are also common. Structural rearrangement and copy number signatures show that whole genome duplication, aneuploidy and complex rearrangements are common. Complex rearrangements occur recurrently and are associated with amplification of *TERT*, *CDK4*, *MDM2*, *CCND1*, *PAK1* and *GAB2*, indicating potential therapeutic options.

[1] QIMR Berghofer Medical Research Institute, Brisbane, QLD, Australia. [2] Melanoma Institute Australia, The University of Sydney, Sydney, NSW, Australia. [3] School of Medicine, The University of Queensland, Brisbane, QLD, Australia. [4] Center for Rare Melanomas, University of Colorado Cancer Center, Aurora, Colorado, USA. [5] Sydney Medical School, The University of Sydney, Sydney, NSW, Australia. [6] Royal Prince Alfred Hospital, Sydney, NSW, Australia. [7] Royal North Shore Hospital, Sydney, NSW, Australia. [8] New South Wales Health Pathology, Sydney, NSW, Australia. [9] Dermatology Clinic, University Hospital Zürich, University of Zurich, Zurich, Switzerland. [10] John Curtin School of Medical Research, Australian National University, Canberra, ACT, Australia. [11] Centre for Cancer Research, Westmead Institute for Medical Research, The University of Sydney, Westmead, Sydney, NSW, Australia. [12] These authors contributed equally: Nicola Waddell, Nicholas K. Hayward. ✉email: felicity.newell@qimrberghofer.edu.au

Acral melanoma (AM) is a rare subtype of melanoma that occurs on non-hair-bearing glabrous skin on the palms, soles and nail apparatus (subungual). In European-derived populations 2–3% of melanoma cases are acral, however AM is the most common subtype in Asian and African populations[1,2]. Compared with cutaneous melanoma (CM), AM has a poorer prognosis, potentially due to diagnosis at a more advanced clinical stage, or due to biological differences favouring tumor aggression[1,3–6].

Next generation sequencing analyses of AM have either involved targeted or exome sequencing[7–9], or whole-genome sequencing (WGS) of small numbers (35 or fewer) of tumors[10–13]. These studies have shown that AM is distinctive, with a lower number of SNV/indel mutations and higher numbers of structural rearrangement variants (SVs) and focal copy number events than in CM. The high mutation burden in CM is attributed to the effect of ultraviolet radiation (UVR) and while not considered a major driver in AM we[13,14] and others[7,9] have shown that small numbers of tumors do exhibit effects of UVR. Mutations in *BRAF* and *NRAS* occur in AM, but at lower rates than in CM[15,16]. Mutations in *NF1* and *KIT*, as well as oncogenic amplification of genes including *CCND1, PAK1, GAB2, CDK4* and *TERT*, are more common events in AM than in CM[7,8,13].

In this work, as the whole-genome landscape of AM is not completely understood, we extend our previous analysis of AM[13] to fully define the key genomic aberrations, and the mutational processes driving this rare melanoma subtype.

## Results

The study involved WGS of 87 fresh-frozen tumor specimens and matched germline DNA (Supplementary Data 1 and 2). Tumor samples were sequenced to a median coverage of 59X (range 35–118) and germline blood samples to a median coverage of 35X (range 21–124). Sixty-three tumors underwent RNA sequencing. Patients had a mean age of 68 and 52% were female. Fifty-nine tumors were from the sole of the foot, six were from the palm of the hand and there were twenty-two subungual tumors (15 toenail, 7 thumbnail/fingernail). Thirty-six tumors were from primary sites and the rest were recurrent (3) or metastatic tumors (48), including a cell line derived from a metastasis. For most patients (75%, 65), there was no treatment listed (either before or after sample collection) in the available clinical data. Males had a lower age at diagnosis of the primary tumor (Supplementary Fig. 1a). More tumors with a higher T classification (T3 or T4) had ulceration (T3 or T4, 64% and T1 or T2 19%, Fisher's exact test, $p = 0.0017$, Supplementary Data 1). As expected, higher N classification (Supplementary Fig. 1b), and locoregional and distant metastasis status of the patient (Supplementary Fig. 1c) were associated with poorer melanoma-specific survival.

**Mutational and rearrangement burden in acral melanoma.** Overall, the SNV/indel mutation burden (tumor mutational burden, TMB) was low, with a median burden of 2.1 mutations per megabase (range of 0.68–34.9) (Fig. 1a). Rearrangements (SVs) were frequent, with a median of 283 rearrangements per tumor (range = 32–1251), and the percentage of the genome affected by copy number variation such as amplification with copy number (CN) $\geq 6$, copy-neutral LOH, copy number loss (CN1) and deletion (CN0), was variable (Fig. 1a), with a median of 26% of the genome affected (range 7–45%). Whole-genome duplication (WGD) was present in 71% (62/87) of tumors (Fig. 1a, which is higher than the average of 28% (PCAWG)[17] to 37% (TCGA)[18] identified across different cancer types, indicating a potential key role for WGD in the tumorigenesis of AM.

The distribution of tumor mutational burden and structural rearrangements in each tumor is shown in Fig. 1b. Thirteen tumors had a higher TMB of $\geq 6$ mutations/Mb (3 times the median of the cohort) in comparison with other tumors, with four having a TMB of >20 mutations/Mb. There was a significant difference in mutation burden based on site of the primary lesion (all tumors, $n = 87$: Kruskal–Wallis test, $p = 8.5 \times 10^{-6}$, Fig. 1c). Subungual tumors from the fingernail/thumbnail had the highest mutation burden and tumors arising on the foot had the lowest mutation burden. Tumors that had undergone WGD had a significantly higher number of mutations (Mann–Whitney U-test, $p = 0.0045$, Fig. 1d). Higher rearrangement count was observed in tumors where the primary tumor was thicker (Mann–Whitney U-test, $p = 0.0007$, Fig. 1e) and therefore those with higher T classification (Mann–Whitney U-test, $p = 0.0042$, Fig. 1f).

**The UVR mutational signature is present in higher TMB tumors.** To investigate the variability in TMB, mutational signature analysis of single base substitutions (SBS), doublet based substitutions (DBS) and indels (ID) was performed (Fig. 2a). UVR related signatures (SBS7a, SBS7b, SBS7c, SBS7d, DBS1 and ID13) were identified in a subset of tumors, with 11 tumors having a greater than 50% contribution of the SBS UVR signatures (Fig. 2b). For tumors with a subungual primary site, 6/7 fingernail/thumbnail tumors and 5/15 toenail tumors had evidence of a UVR signature. UVR signature was significantly associated with higher TMB (Kruskal–Wallis test, $p = 3.5 \times 10^{-5}$, Fig. 2c). Other SBS signatures were identified in single tumors with a higher TMB (Fig. 2a, b), including one tumor (MELA_0271) with a mismatch repair signature (SBS21/SBS26 and ID2) due to a somatic homozygous deletion (copy number 0) affecting *MLH1*, and another tumor (MELA_0015) with a > 30% contribution of signature 17a/b, a signature we have also previously identified in mucosal melanoma (MM)[19]. Another tumor with a higher TMB within the cohort (MELA_0870) had evidence of SBS32, a signature found in cutaneous squamous cell carcinoma in association with prior treatment with azathioprine to induce immunosuppression[20,21], however, this patient had no history of azathioprine treatment.

Signatures 1, 5, and 40, which are associated with aging and found in most cancer types[20], were present in the majority of tumors. Other signatures were also present in tumors with a lower TMB. SBS38, postulated to be the result of indirect UVR[20], was present with a > 25% contribution in 9 tumors and these tumors had no contribution of a UVR signature. The etiology of SBS38 may be associated with oxidative stress as the signature was reported to cluster with other oxidation damage repair related signatures when using the simple probabilistic model DNA Repair FootPrint (*RePrint*) to represent each signature[22]. APOBEC (SBS2, SBS13) signatures were present at low levels ($\leq 30\%$ contribution) in 51 tumors, and these had a higher number of rearrangements (Fig. 2d). One tumor (MELA_0872) had cisplatin signatures SBS31 and DBS5 and was a subcutaneous metastasis collected after the patient had undergone cisplatin chemotherapy. Indel signatures for non-UVR tumors were ID1, ID2 and ID8, which correlate with age of diagnosis, and ID9, which has no proposed etiology but is found in many cancer types[20].

**Rearrangement and copy number signatures.** Rearrangement signatures (RS) and copy number signatures (CNS) were extracted to further understand the associations of rearrangements and copy number events with WGD, aneuploidy and complex rearrangements. Three RSs were identified in the cohort and had strong cosine similarity to the signatures previously identified in breast cancer[23] (Fig. 3a, Supplementary Fig. 2a, b). Signatures RS4 and RS6 are clustered, RS2 is not clustered. We previously observed these clustered signatures (RS4/RS6) in a

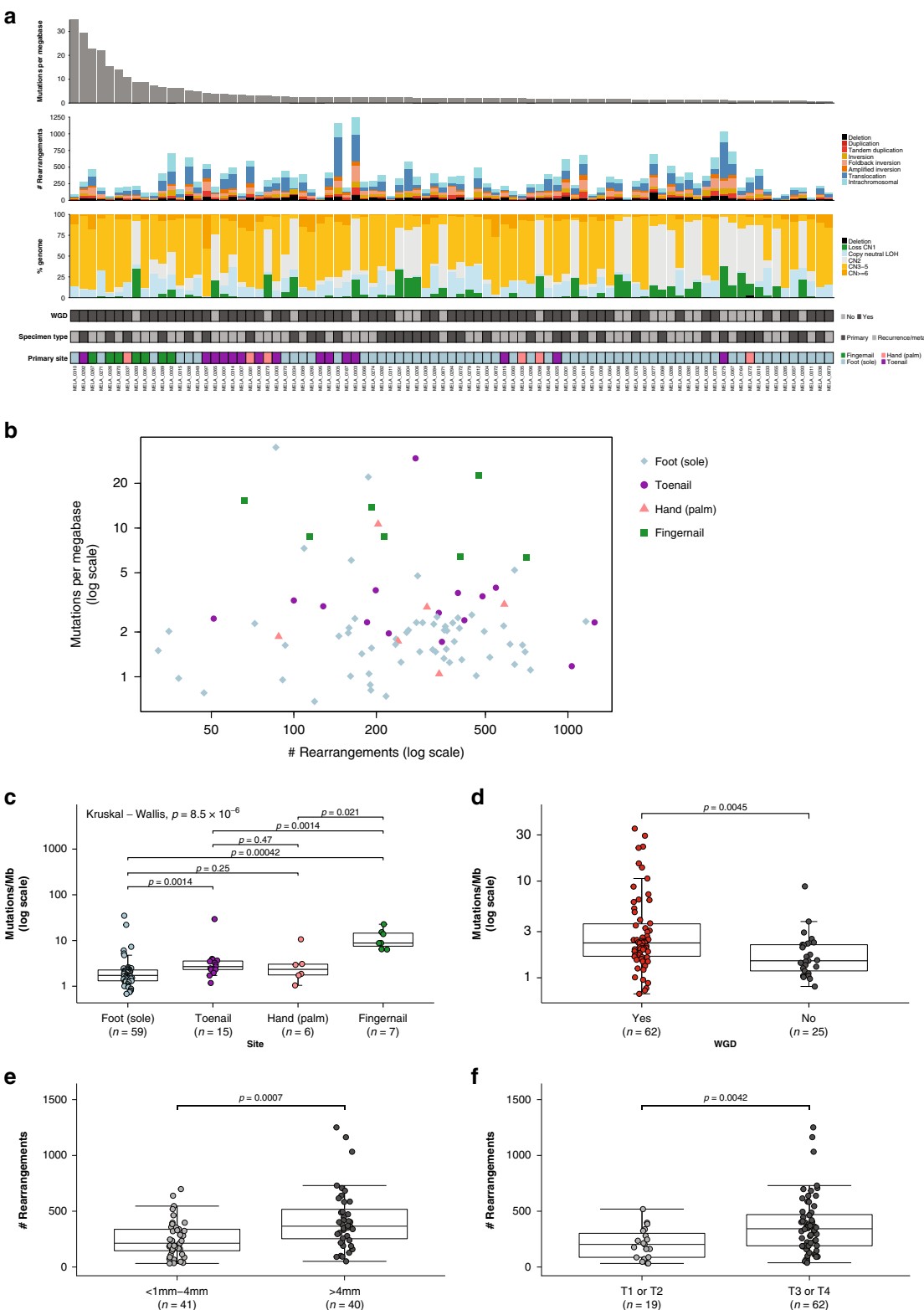

subset of MMs with evidence of localized rearrangements[19]. We extracted five copy number signatures which were similar to signatures identified in sarcomas[24]: CNS1, CNS3, CNS5, CNS6, CNS7 (Fig. 3a, Supplementary Fig. 2c, d).

Most samples with high amounts of non-clustered RS2 (>50%) had undergone WGD (84%, 21/25) and only three had evidence of CNS5. Tumors with diploid genomes had a high contribution of CNS3, whereas CNS1, CNS6, CNS7 occurred primarily in

tumors with WGD (Fig. 3b). There were more subungual tumors with any evidence of CNS1 (59% of subungual tumors) than tumors from the palms of hands or soles of feet (23%, Fisher's exact test, p = 0.003). CNS6 had weaker cosine similarity with the sarcoma signature CNS6 and the proportion of CNS6 in tumors was negatively correlated with the number of rearrangements (Fig. 3c), indicating that this signature is associated with less genomically rearranged tumors. In sarcomas, CNS5 was

**Fig. 1 Somatic variant burden. a** From top to bottom: mutations per megabase (where mutations includes single nucleotide (SNV), dinucleotide (DNV) and trinucleotide variants (TNV) and indels (small insertions and deletions); number and type of structural rearrangement variants; percent of the genome affected by copy number aberrations; whether a tumor has undergone whole genome doubling (WGD); specimen type (primary or recurrence/ metastasis); site of primary tumor. **b** Scatterplot of mutations per megabase (log scale) versus structural rearrangement count (log scale) with points colored by site. **c** Box plot and overlaid scatterplot of mutation burden (SNV,DNV,TNV, small indel) across different primary sites. Kruskal–Wallis test was used to determine overall significance between signatures and pairwise Mann–Whitney U-tests to compare each pair of sites. The pairwise test p-values displayed are adjusted p-values after correction for multiple testing by FDR. **d** Box plot of number of mutations per megabase with or without whole genome doubling (Mann–Whitney U-test). **e** Box plot of number of rearrangements in samples with primary tumor thickness of >4 mm or <1–4 mm (Mann–Whitney U-test). **f** Box plot of number of rearrangements in samples with primary T Classification of T1 or T2 compared with T3 or T4 (Mann–Whitney U-test). In each box plot, the box boundaries show the first to third quartiles, the median is the center line and the whiskers represent 1.5 times the inter-quartile range.

associated with chromothripsis and in agreement with this, the proportion of CNS5 in AM was positively correlated with the number of rearrangements (Fig. 3d) and proportion of clustered rearrangements (RS4 and RS6) (Fig. 3e). There was no difference in the contribution of CNS5 in tumors with and without WGD in AM (Fig. 3b), indicating that clustered complex rearrangements are equally likely to occur in both diploid and WGD tumors.

**WGD, aneuploidy and localized structural rearrangements.** Within the cohort, copy number amplifications and deletions, rearrangements and aneuploidy, as well as regions of hypermutation (kataegis), were highly recurrent across the genome (Fig. 4a), as suggested by rearrangement and copy number signatures and confirming findings from previous studies[10,11,13,15]. Aneuploidy, the gain or loss of whole chromosomes or chromosome arms (where gain or loss was considered to be above or below the ploidy of the respective tumor), was common with a number of chromosome arms affected in >30% of samples (Fig. 4a, b). Some events affected both arms of a chromosome: gain of chromosomes 7 (32% of tumors) and 8 (28%) and loss of chromosomes 9 (37%) and 10 (47%). Tumors with 6p gain and 6q loss often had both events (38% of tumors), potentially resulting in an isochromosome, and was more common in subungual tumors (68% subungual compared with 28% other acral sites, Fisher's exact $p = 0.002$). Signatures CNS1 and CNS6 were positively correlated with aneuploidy (Fig. 4b, Supplementary Fig. 3a). Subungual melanomas, which have high amounts of CNS1, were also more aneuploid when compared with other tumors (Mann–Whitney U-test, $p = 0.0012$, Supplementary Fig. 3b). Tumor aneuploidy was significantly higher in tumors with WGD (Mann–Whitney U-test, $p = 2.6 \times 10^{-10}$, Fig. 4b, Supplementary Fig. 3c), supporting a previous report[25] that higher ploidy tumors are more prone to aneuploidy.

Complex rearrangements (breakage-fusion bridge (BFB), chromothripsis, or localized complex for those not fitting criteria for BFB or chromothripsis) were common, with 84% of tumors having at least one chromosome with evidence of localized complexity (Fig. 4c). Many tumors showed evidence of BFB, and complex events were equally common in diploid or WGD tumors (Fisher's exact $p = 0.33$). Longer melanoma-specific survival and decreased risk of melanoma specific mortality was associated with the presence of complex chromosomes in a tumor (log-rank test, $p = 0.028$, Supplementary Fig. 4a, multivariable Cox survival model, $0 = 0.006$, Supplementary Fig. 4b). The number of chromosomes harboring localized complex rearrangements had a positive correlation with the proportion of CNS5 (Supplementary Fig. 4c). CNS5 has been associated with chromothripsis[24] and therefore the signature characterizes the effects of complex rearrangements on copy number, including retention of heterozygosity and segments of high level amplifications. Kataegis loci were positively correlated with rearrangement burden (Supplementary Fig. 4d) and SNVs that fell in kataegis loci had an

APOBEC signature (Supplementary Fig. 4e), agreeing with reports in other cancers, including MM[19,26].

Recurrent complex rearrangements were observed on chromosomes 5, 6, 7,11 and 12 and were most frequent on chromosomes 11 (43% of tumors) and 5 (34%). Locations targeted (Supplementary Fig. 5) were the start of 5p, in the region where TERT is located (Supplementary Fig. 5a), the start of chromosome 11p in the area with CCND1, PAK1 and GAB2 (Supplementary Fig. 5b), and on chromosome 12p including genes CDK4 and MDM2 (Supplementary Fig. 5c). For complex events on chromosome 6 (Supplementary Fig. 5d) either the entire p or q arm was often affected, whereas chromosome 7 events (Supplementary Fig. 5e) occurred primarily on the p arm (11/16), concentrated towards the telomere. Recurrent translocations (tumors with ≥5 linking translocations), were observed between those chromosomes with localized complex events (Fig. 4c, Supplementary Fig. 6a, b), occurring on chromosome 6 and 11 (15 tumors, 17%, Supplementary Fig. 6c); chromosomes 5 and 7 (9 tumors, 10%); chromosomes 5 and 11 (8 tumors, 9%), and chromosomes 5 and 12 (8 tumors, 9%).

**Significantly mutated gene analysis.** Significantly mutated gene (SMG) analysis identified six genes: BRAF (20 tumors), NRAS (16), NF1 (10), TYRP1 (7), PTEN (6) and NOTCH2 (4) (Supplementary Fig. 7a, Supplementary Data 3 and 4). No new significantly mutated genes were identified when analyzing subgroups of primary, recurrence/metastasis tumors, subungual tumors or tumors from the sole of the foot or palm of the (Supplementary Data 4). BRAF mutations were all missense, with 16 at the p.V600E hotspot. Only one subungual melanoma had a BRAF mutation (toenail with p.V600E). BRAF mutant tumors were found in patients of younger age at diagnosis of primary (Mann–Whitney U-test, p = 0.021, Supplementary Fig. 7b). BRAF mutant tumors were also associated with lower primary T classification (52% of T1 or T2 tumors were BRAF mutant compared with 16% of T3 or T4 tumors, Fisher's exact $p = 0.0044$). NRAS mutations were missense in the p.Q61 (11 tumors) or p.G12 hotspots (5). BRAF and NRAS hotspot mutations were mutually exclusive (Fisher's exact, $p = 0.018$). All NOTCH2 mutations and most NF1 mutations were putatively protein truncating loss of function (LoF). PTEN mutations were often frameshift indels (4/6) and were not present in subungual melanomas.

TYRP1, which encodes an enzyme involved in the generation of eumelanin in melanocytes, was affected by frame-shift indels in 7 tumors. The variant was found to be expressed in 3/5 tumors that had matching RNA-seq. TYRP1 mutations were from tumors with a primary site of the foot with 3/7 being subungual toenail and 6/7 mutations were in recurrence/metastasis specimens. Six of the seven mutations were the same variant, a frame-shift indel at p.N353Vfs*31 (Supplementary Fig. 7c). This variant is present at a low frequency in the Gnomad database[27] (allele frequency of $8.2 \times 10^{-5}$) and has been reported as pathogenic in Clinvar (rs387906562) for oculocutaneous albinism type 3 (OCA3), an

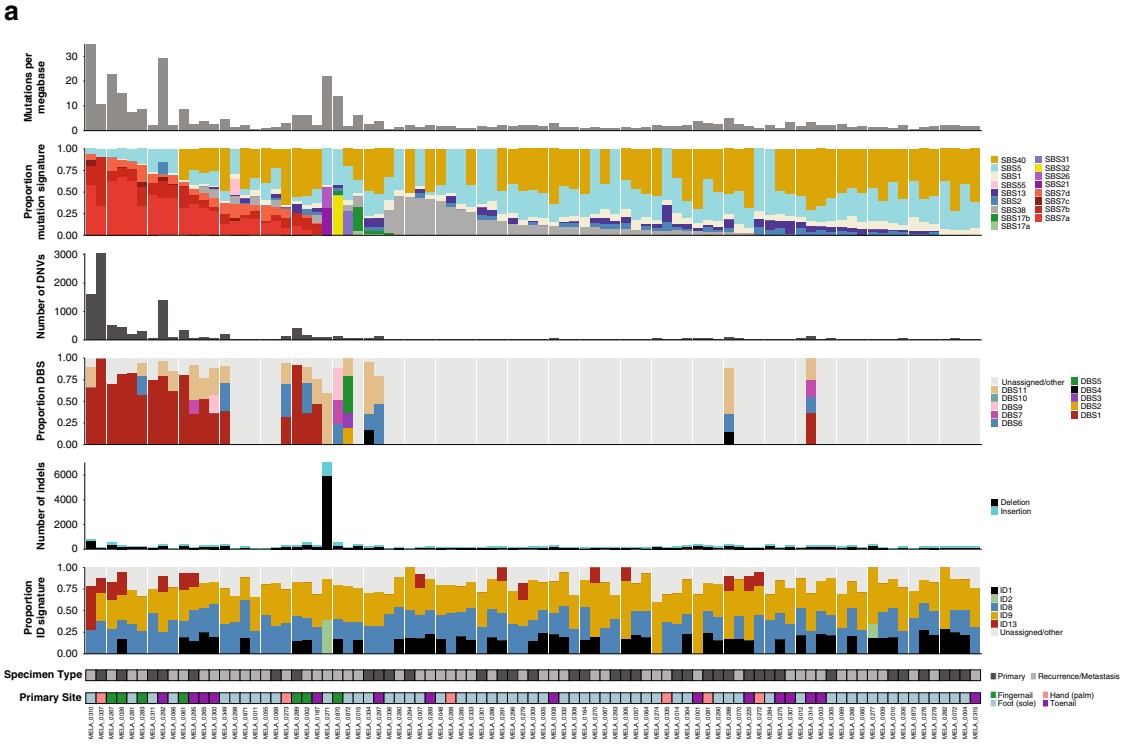

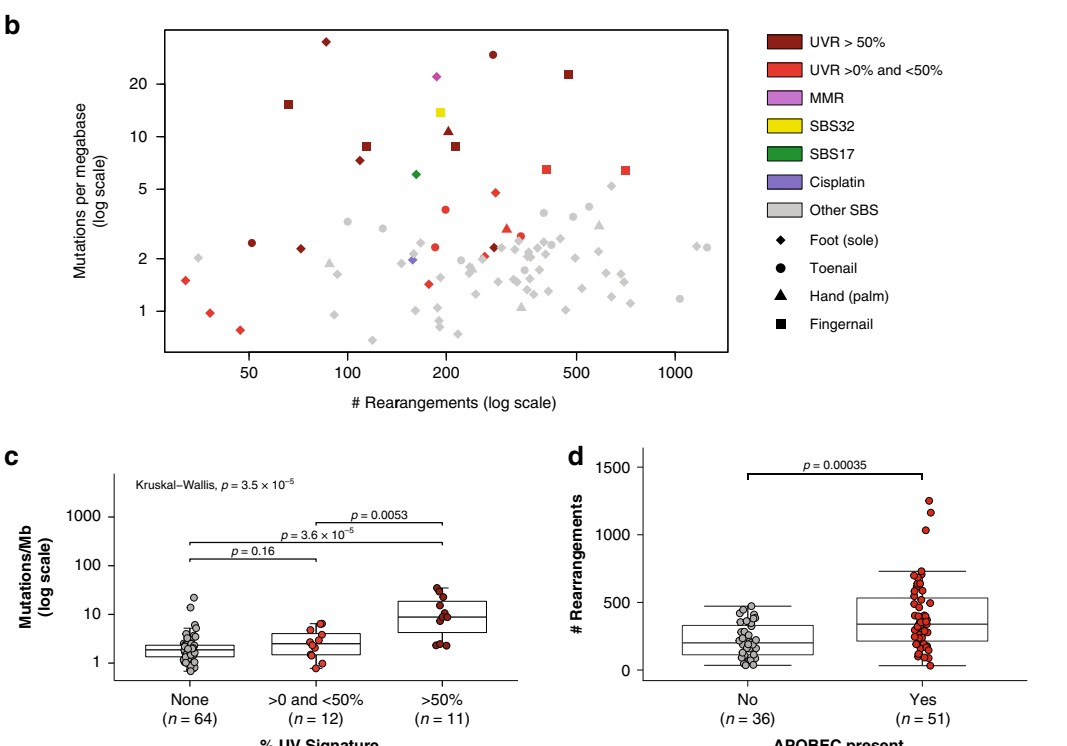

**Fig. 2 Mutational signatures of point mutations, dinucleotide mutations and indels. a** From top to bottom: mutations per megabase, proportion of SBS mutational signatures; number of DNVs; proportion of DNV signatures (DBS); number of indels; proportion of indel signatures (ID), specimen type (primary or recurrence/metastasis); site of primary tumor. **b** Scatterplot of mutations per megabase versus structural rearrangement count with points colored by SBS signatures and shape indicating different primary sites. **c** Box plot of mutation burden (SNV, DNV, TNV, small indel) across samples with different proportions of UVR signature. Kruskal–Wallis test was used to determine overall significance between the groups and pairwise Mann–Whitney *U*-tests to compare each pair. The pairwise test *p*-values displayed are adjusted *p*-values after correction for multiple testing by FDR. **d** Box plot of number of rearrangements in samples that have evidence of an APOBEC signature or have no evidence of the signature (Mann–Whitney *U*-test). In each box plot, the box boundaries show the first to third quartiles, the median is the center line and the whiskers represent 1.5 times the inter-quartile range.

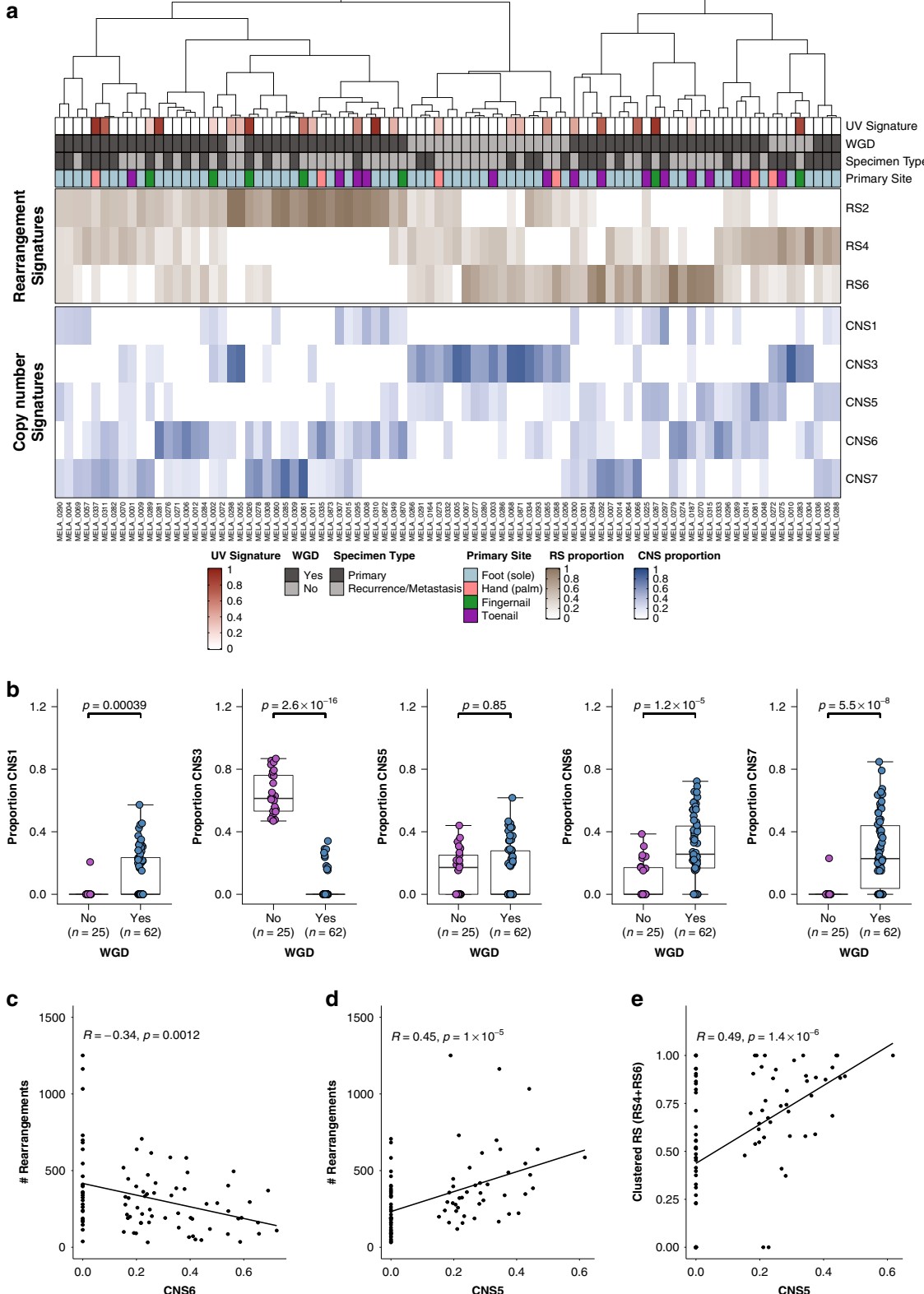

**Fig. 3 Rearrangement and copy number signatures. a** Unsupervised hierarchical clustering of rearrangements signatures (RS2, RS4, RS6) and copy number signatures (CNS1, CNS3, CNS5, CNS6, CNS7). **b** Box plots of the proportions of (from left to right) CNS1, CNS3, CNS5, CNS6, CNS7 in samples which have undergone whole genome duplication and those which have not. In each box plot, the box boundaries show the first to third quartiles, the median is the center line and the whiskers represent 1.5 times the inter-quartile range. **c** Pearson's correlation of CNV signature CNS6 with numbers of rearrangements. **d** Pearson's correlation of CNV signature CNS5 with numbers of rearrangements. **e** Pearson's correlation of CNV signature CNS5 with proportion of clustered rearrangement signatures (RS4 and RS6).

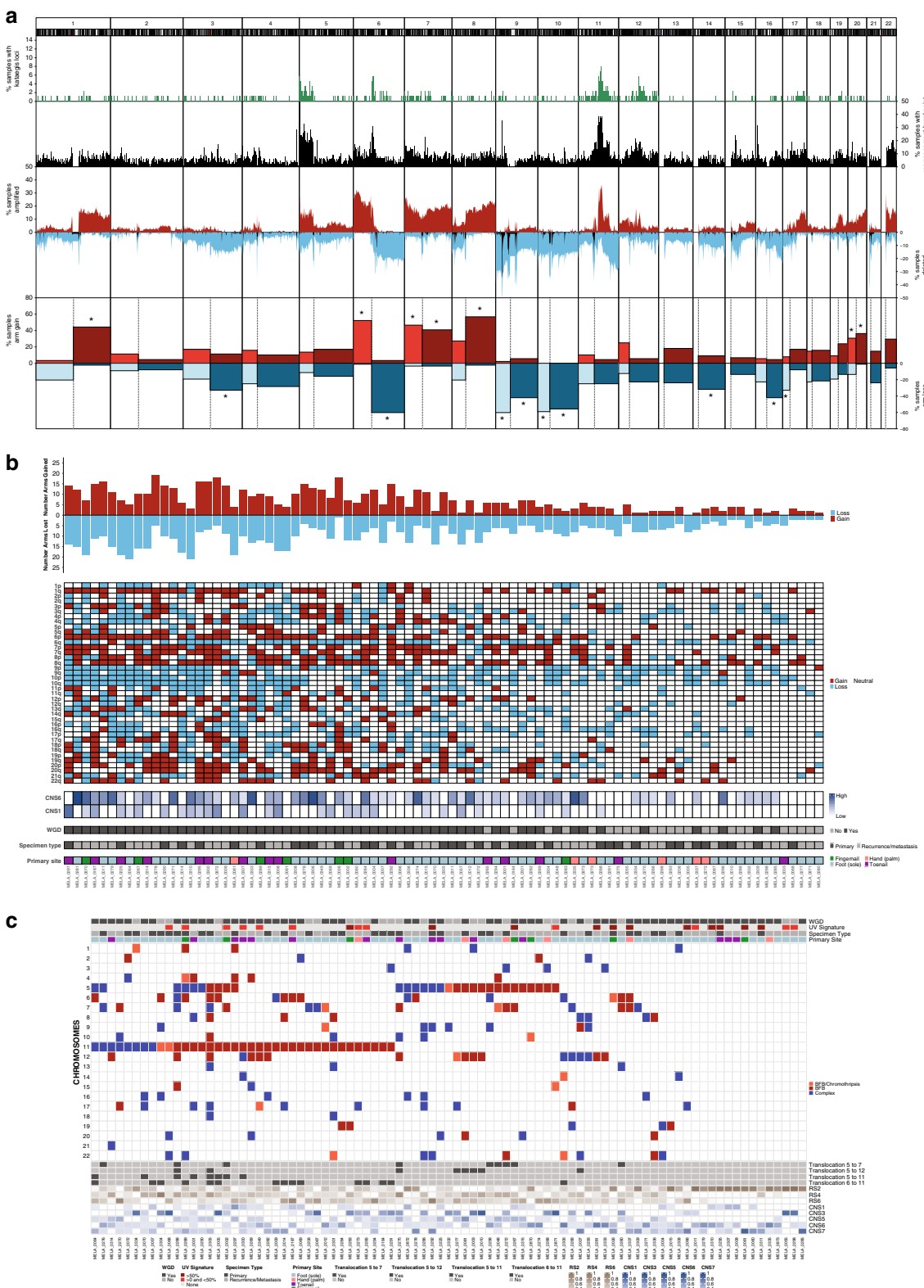

**Fig. 4 Genomic complexity and chromosomal instability including aneuploidy and localized rearrangement events. a** Genomic summary of numerical and structural instability in 1 Mb regions across the genome. From top to bottom: % samples with kataegis loci (green), % samples with rearrangement breakpoints (black), % samples with amplification (CN ≥ 6, red), % samples with loss or deletion (CN0 or CN1 in blue) and % samples with deletion (CN0 only in black), % samples with chromosome arm gain (red) or loss (blue). Arms with more than 30% of samples affected are indicated with an asterisk. **b** Whole arm chromosomal gains or losses in each sample. For acrocentric samples, only the p arm is shown (chromosomes 13, 14, 15, 21, 22). The number of chromosome arms with a gain (red) or loss (blue). **c** Chromosomes with localized complex rearrangements including breakage-fusion-bridge and chromothripsis.

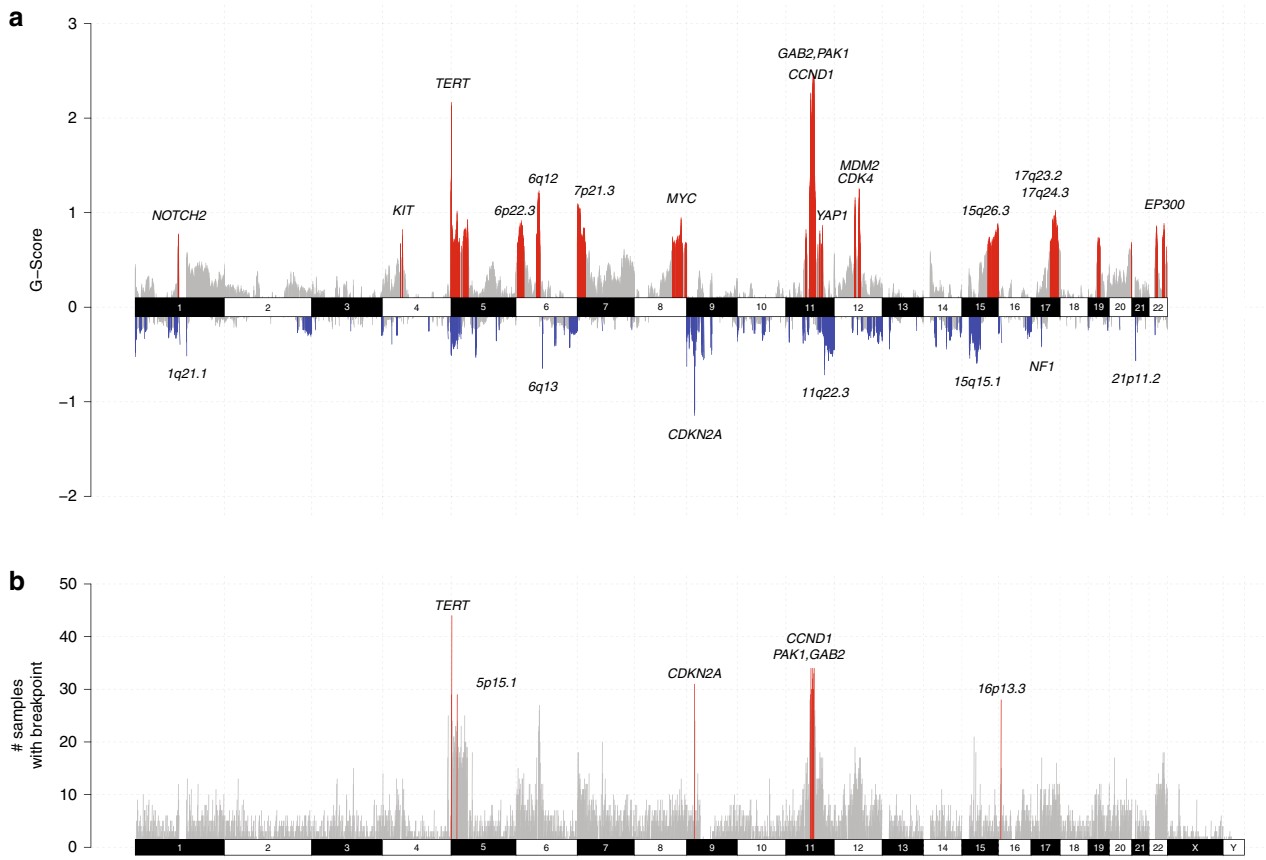

**Fig. 5 Recurrent focal regions of copy number and rearrangement breakpoints. a** Focal regions of recurrent amplification (red) and deletion (blue) as identified by GISTIC2. Genes and chromosomal cytobands of interest are annotated in the plot. **b** Regions of recurrent rearrangement breakpoints as identified by RETREAD. The plot shows 1 Mb windows that have 1 or more rearrangements. Bars in gray represent windows that are not significant ($q >$ 0.2), and red bars indicate regions that are significantly enriched ($q < 0.2$).

autosomal recessive disorder of melanin biosynthesis that reduces pigmentation of the hair, skin and eyes[28]. Mutations in OCA genes including *TYRP1* have been proposed to confer a moderate risk for CM[29]. The p.N353Vfs*31 variant was not found in a previous analysis of 38 exome-sequenced AM specimens[7], and p. N353Vfs*31 was found rarely (17/46651 tumors) in other cancer types queried in cBioportal[30,31].

Mutations occurred in other oncogenes, including *KIT* (9), *MAP2K1* (2), *KRAS* (2) and *HRAS* (1). *KIT* mutations were primarily missense mutations (8/9) that occurred within the kinase domain of the protein. Mutations (including putatively LoF) in other tumor suppressors occurred in small numbers of tumors (Supplementary Fig. 7a). *TERT* promoter mutations (9) were rarer than in CM[13] with four occurring at position −124 (C228T).

**Recurrent rearrangements and copy number aberrations.** Recurrent regions of focal copy number loss, as determined using GISTIC2 (Fig. 5a), included those containing *CDKN2A* (9p21.3) and *NF1* (17q11.2). The region containing *PTEN* was not a significant region of loss, but homozygous deletions were found in 3 tumors. Significantly amplified regions included those that were recurrently affected by complex genomic rearrangements, and these regions harbored genes including *CCND1* and *GAB2* (and close to *PAK1*), *TERT*, *YAP1*, *MDM2*, *CDK4*, *NOTCH2*, *KIT*, and *EP300*. Broadly similar patterns were observed when analyzing foot/hand (Supplementary Fig. 8a) or subungual subgroups (Supplementary Fig. 8b). Subungual tumors were significantly amplified in the regions 15q26.3 and the region on chromosome 4 including *KIT* whereas non-subungual foot/hand tumors were

not. Of 7 tumors with *KIT* amplifications, 4 were subungual. Similar focal amplification and deletion regions were observed between primary tumor (Supplementary Fig. 8c) and recurrence/ metastasis tumor subgroups (Supplementary Fig. 8d). However, focal amplifications on chromosome 22p (including *EP300*) were only significant in primary tumors, with *EP300* amplifications occurring more often in primary tumors (31%) than recurrence/ metastasis tumors (8%) (Fisher's exact test, $p = 0.0087$).

Recurrent rearrangement regions included those affected by recurrent localized complex events and copy number amplifications: a 3 Mb region at cytoband 5p15.33 (*TERT*), regions on chr11 cytobands 11q.14.1 and 11q13.3 (*CCND1, GAB2, PAK1*), *9p21.3 and 16p13.3* (Fig. 5b). Cytoband 9p21.3 encompasses *CDKN2A* which has predicted LoF rearrangements. The cytoband 16p13.3 contains the gene *RBFOX1*, a large gene that is reported to resemble common fragile sites[32]. Although not significant, a high number of breakpoints was present on 15q13.3-15q14. The genes most affected by breakpoints were *GREM1, FMN1, RYR3*. However *SPRED1*, a Ras-MAPK pathway gene in the region previously reported to be mutated in AM[8] and MM[19,33] and reported to act as a tumor suppressor in MM[33] was also affected, with 14 tumors having predicted LoF rearrangement events. *NF1* rearrangements occurred in 14 patients, with predicted LoF for 11, extending the total number of somatic *NF1* LoF aberrations to 18 (21% of tumors, 7 SNV/indel, 11 SV). Eight predicted LoF rearrangements in *PTEN* were also identified.

Foot/hand (Supplementary Fig. 9a) and subungual (Supplementary Fig. 9b) subgroups were analyzed separately. No significant regions were identified in subungual tumors, likely

due to the small numbers analyzed ($n = 22$), but the pattern of rearrangements was broadly similar when compared with non-subungual tumors from the foot and hand. *SPRED1* rearrangements were more common in subungual tumors (32%) than in tumors from non-subungual sites (11%, Fisher's exact test, $p = 0.039$). When comparing primary (Supplementary Fig. 9c) and recurrence/metastasis tumor (Supplementary Fig. 9d) subgroups, the overall distribution of rearrangement breakpoints was mostly consistent. A new recurrent region of rearrangement breakpoints was identified in primary tumors on 4q34.3. This region contains a long non-coding RNA, *LINC00290*, which has been reported to be a recurrent deletion site in a pan-cancer study[18] and has been suggested as a common fragile site[34]. Breakpoints in *RBFOX1* (16p13.3) were also more common in primary tumors (50%) than recurrence/metastasis tumors (20%, Fisher's exact test, $p = 0.005$).

In total, 3.1% (887/28422) of the rearrangements resulted in in-frame gene fusions with predicted correct orientation of genes and phased exons (Supplementary Data 5). Recurrently fused kinase genes included *TRIO* (11 fusions in 3 tumors), *PAK1* (9 fusions in 6 tumors), *DGKB* (7 fusions in 4 tumors), and *DCLK1* (3 fusions in 2 tumors). Most genes involved in recurrent fusion events occurred in regions with complex rearrangements and/or contained multiple other breakpoints within the genes making it difficult to accurately predict the true functional consequence of these fusions. One *BRAF* fusion (*GTF2IRD1-BRAF*) that retained the *BRAF* kinase domain was identified, however 3 other *BRAF* breakpoints were also present in the same sample. One *GOPC-ROS1* fusion was also identified, which has previously been described in this patient in a case report describing response to the tropomyosin receptor kinase (TRK) inhibitor entrectinib[35].

**Impact of somatic aberrations on gene expression**. When combining all somatic aberrations in candidate driver genes, lower expression was observed in most genes with putative LoF mutations in tumors with transcriptome sequencing (Supplementary Fig. 10a) and higher expression with putatively activating mutations (Supplementary Fig. 10b). The exceptions were *NRAS* and *CDK6* (both with only a trend for higher expression), *TYRP1*, and *TERT*. Upstream breakpoints have been reported to increase expression of *TERT* due to mechanisms such as enhancer hijacking in other cancer types[36,37] and are also proposed to occur in AM[7]. Breakpoints within the region 20 kb upstream of *TERT* were identified in 25 tumors, and 14/25 had no other *TERT* aberration (Supplementary Fig. 10c). For 22/25 upstream breakpoints, the translocation partner of the breakpoint was within 100 kb of a super enhancer (SE) that is defined in the dbSUPER database[38] or in melanoma cell lines in the SEdb database[39]. Seven of the ten tumors with the highest *TERT* expression had upstream breakpoints, suggesting that expression of *TERT* may potentially be transcriptionally activated by super enhancers in close proximity to *TERT*. In contrast with MM[19], no association with telomere length was found in AMs with *TERT* mutations (Supplementary Fig. 10c).

**Summary of genomic events in acral melanoma**. A summary of somatic aberrations, combining point mutations, indels, copy number aberrations and structural variation is shown in Fig. 6a. *SPRED1* aberrations were more common in primary tumors (Fisher's exact test, $p = 0.022$, 10% of recurrence/metastasis, 31% of primary). *CDK4* aberrations were more common in subungual tumors (Fisher's exact test $p = 0.017$, 32% of subungual, 9% of tumors from other sites) with 6/7 subungual *CDK4* aberrations being from the toenail. There were no associations of total aberrations in a gene with melanoma-specific survival, with the exception of *SPRED1* ($p = 0.044$). Specifically, poorer survival was associated with samples that had a *SPRED1* aberration in

combination with another cancer driver mutation: *NRAS*, *NF1* or *KIT* (log-rank test, $p = 0.0074$, Fig. 6b). In a multivariable Cox regression survival model based on *SPRED1* mutation status, overall stage, age, gender and specimen type (primary or recurrence/metastasis), *SPRED1* aberrations co-occurring with *NRAS/NF1/KIT* mutations were associated with increased risk of melanoma-specific mortality ($p = 0.006$) (Fig. 6c). *PTEN* aberrations in primary tumors were also associated with poorer survival (log rank test, $p = 0.029$, Supplementary Fig. 11a). However, only 5 tumors had *PTEN* aberrations and there was only a trend for increased risk of melanoma specific mortality in a multivariable Cox survival model ($p = 0.052$, Supplementary Fig. 11b).

AMs exhibited many similar driver genes with CM; however after classifying AMs by the genomic subtypes identified in CM by TCGA[40], differences in the overall distribution were observed (Fig. 6a). Compared with CM from our previous whole genome analysis (140 tumors)[13], there were fewer *BRAF* hotspot (18% AM; CM 46%) and RAS hotspot mutations subtype tumors (21% AM; 31% CM) in AM. There was a higher proportion of *NF1* subtype (23% AM, 10% CM) and the Triple Wild Type (Triple WT) tumors represented the highest proportion of genomic subtypes (38% AM, 11% CM). In the acral melanomas, there was a significant association with tumor thickness (Fisher's exact test, $p = 0.0046$) and ulceration (Fisher's exact test, $p = 0.00056$). *BRAF* subtype (81% *BRAF* subtype tumors with Breslow thickness <1 mm–4 mm) and Triple WT tumors (58% Triple WT with Breslow thickness <1 mm–4 mm) were associated with thin to intermediate tumor thickness and thicker tumors were associated with the *NF1* (72% with Breslow thickness >4 mm) and *RAS* subtypes (69% with Breslow thickness >4 mm). The presence of ulceration was associated with *NF1* (present in 77%) and *RAS* (present in 75%) subtypes and more likely to be absent in the *BRAF* subtypes (absent in 87%). The number of rearrangements differed between subtypes (Kruskal–Wallis $p = 0.022$), with the lowest rearrangement counts in *BRAF* (mean 203) compared with *NRAS* (323), *NF1* (350) and Triple WT (374).

A BRAF V600E subclassification for acral melanomas has been previously proposed[8], and in our cohort, these tumors appeared similar to CM, with low rearrangement burden and fewer samples with complex chromosomes (Fisher's exact test, $p = 0.011$). While BRAF V600E mutated tumors were of lower tumor thickness and without ulceration at the time of diagnosis, most BRAF V600 positive tumor samples were recurrence/metastasis specimens (14/16) and were not from subungual sites (1/16 was subungual). *NF1* subtype tumors had higher numbers of tumors with complex events on chromosome 11 (65%, Fisher's exact test = 0.0099) when compared with other subtypes. *CCND1*, *PAK1* and/or *GAB2* mutations were more common in *NF1* (70%) and Triple WT subtypes (52%) and less common in *RAS* (22%) and *BRAF* (6%) subtypes (Fisher's exact test, $p = 0.00016$). Amplifications of other genes such as *BRAF*, *CDK4* and *MDM2* were common in Triple WT subtype tumors as were aberrations in genes that are more rarely mutated in CM such as *KIT* and *SPRED1*.

**Candidate treatment approaches for acral melanoma**. Due to the rarity of AM, less is known about rates of response to immune checkpoint blockade therapy, though one study reported an objective response rate of 32%[41]. High TMB (here defined as the number of SNV and indel mutations per megabase) is a feature associated with response to immunotherapy and although there is no specific cutoff to define a high tumor mutational burden (TMB), a cutoff of ≥20 mutations/Mb has been reported, with this definition for TMB-high used in the Foundation Medicine FoundationOne gene profiling test[42,43]. A subset of four AM had a high TMB (≥20 mutations/Mb), a feature common in

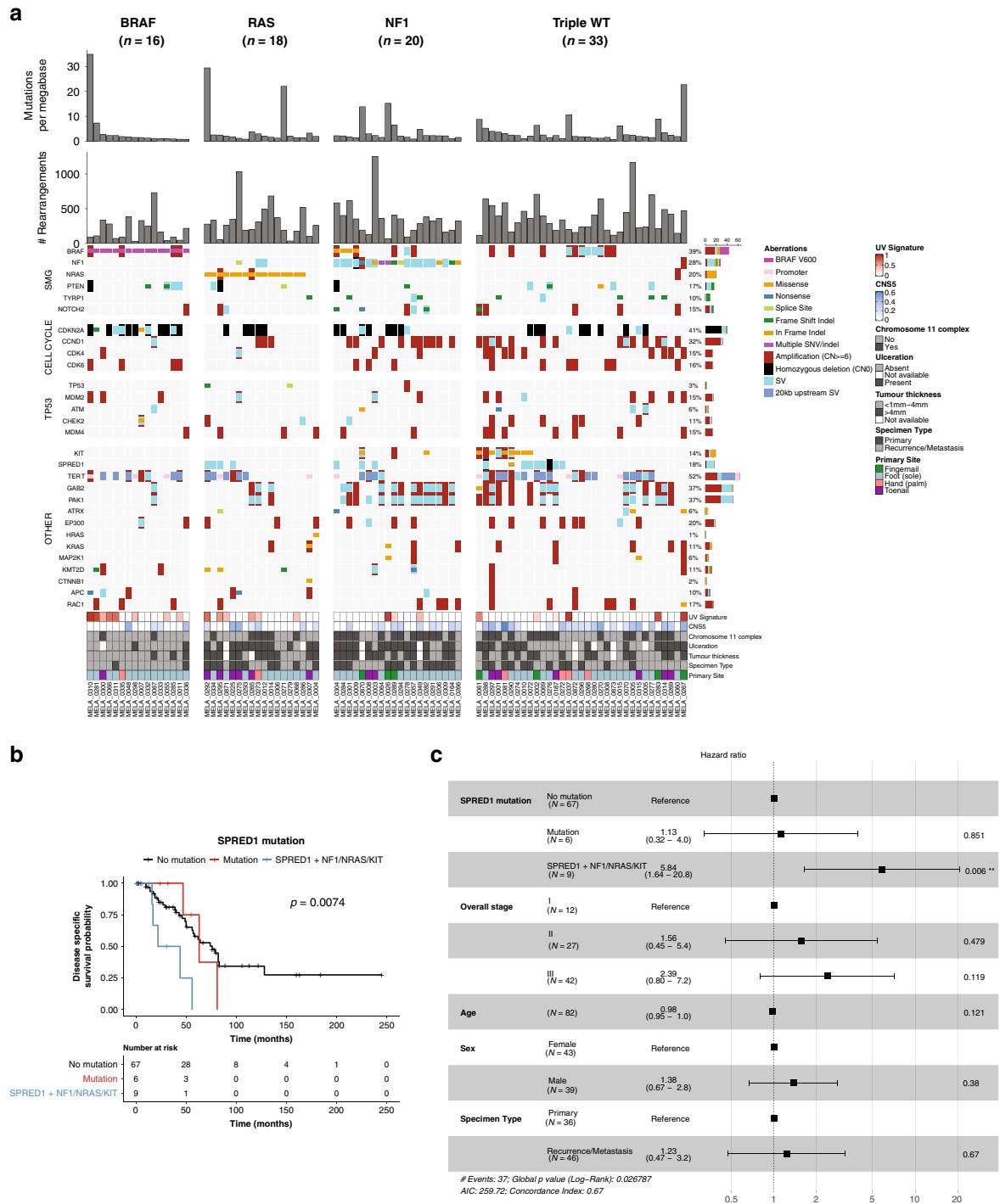

**Fig. 6 Acral subclassifications and somatic aberrations in key acral melanoma genes. a** Overview of mutations divided by TCGA cutaneous melanoma molecular subclassifications of (left to right): BRAF hotspot (V600) mutated tumors; RAS hotspot mutated tumors; *NF1* mutated tumors and Triple wild type (Triple WT). Genes are separated into: significantly mutated genes (from SNV/indel analysis), TP53, and cell cycle pathway genes, with the remaining genes grouped as Other. In the barcharts in the upper panels, each barchart represents data from *n* = 87 tumors, where each bar represents the number of mutations per megabase or number of rearrangements for a single tumor. **b** Kaplan–Meier plot of melanoma-specific survival with log-rank test in patients with or without *SPRED1* mutations and co-occurring mutations in *KIT*, *NRAS* or *NF1*. **c** Forest plot for a multivariable Cox survival model based on *SPRED1* mutations, overall stage, patient age at primary and sex.

immunotherapy responders[44], with a further nine with an intermediate TMB (6–19 mutations/Mb), many of which had specific mutational signatures associated: UVR, MMR, SBS32 and SBS17. In tumors with matching RNA-seq (*n* = 63), the number of expressed neoantigens was significantly associated with higher TMB (Supplementary Fig. 12a, Supplementary Fig. 12b). Other

markers of immunotherapy response: expression of PD-L1 (Supplementary Fig. 12c) and the proportion of CD8+ T cells (Supplementary Fig. 12d) estimated from immune cell deconvolution using CIBERSORT were variable across the cohort. Of the TMB-high tumors, all were above the median. Therefore a subset of AM may be more likely to respond to immunotherapy.

Somatic aberrations identified in AM offer a range of other treatment options. The BRAF V600E subgroup[8] are more similar to CM and targeted treatment with BRAF inhibitors may therefore have utility. *CDK4*, *CDK6* and *CCND1* amplifications were present in 53% of tumors, predominantly in the *NF1* and Triple WT subtypes. AM cell lines and PDX with CDK4 pathway aberrations have been reported to be sensitive to CDK4/6 inhibitors[45], indicating these drugs may have utility in a large proportion of AMs. Other potential treatments for tumors, particularly in the Triple WT subtype subgroup include *KIT* inhibitors, PARP inhibitors for *ATM* LoF mutations[46], and in the one tumor with a ROS1 fusion, treatment with the tropomyosin receptor kinase inhibitor entrectinib[35].

## Discussion

In the present study, we have described the largest whole-genome analysis of acral melanoma to date. The use of mutational signatures including substitution, indel, structural rearrangement and copy number signatures provides important insights into the mutational processes at play in AM. We demonstrated that SBS38, postulated to be the result of indirect UVR damage[20], in fact occurs almost exclusively in tumors that have no UVR signature, indicating an independent cause, potentially oxidative stress[22]. We identified a previously under-appreciated role for UVR in the genesis of AM, in particular with respect to AM arising in subungual sites, indicating that the nail apparatus is insufficient for protection from the effects of UVR. These tumors also appeared to be somewhat genomically distinct from other acral sites. Although not confined to a single molecular subtype, subungual tumors lacked *BRAF* and *PTEN* mutations and were more likely to have *SPRED1* rearrangements or *CDK4* aberrations. Subungual tumors were characterized by higher numbers of mutations per megabase, and were more aneuploid than those in other sites, harboring a higher proportion of CNS1 and chromosome 6 isochromosomes. Cases of subungual melanoma were a minority within the study and predominantly of European ancestry. It would therefore be of interest to explore the UVR signature and other genomic features of subungual tumors in a series of tumors from different ethnicities. The genomic features of primary and recurrence/metastasis tumors were broadly similar, although some aberrations, including *EP300* amplifications, *SPRED1* aberrations and rearrangements in the regions of *RBFOX1* and *LINC00290* were more common in primary tumors. Given the small sample size of primary tumors ($n = 36$), a study comparing larger cohorts of primary and recurrence/metastasis tumors would be of interest to further understand any differences.

The different mutational signatures provided insights into the genomic complexity of acral melanoma, with WGD in the majority of tumors, a high degree of aneuploidy, and recurrent complex rearrangements. The relationship and timing of chromosomal instability and complex genomic rearrangements is of interest. In many cancers, whole genome doubling is thought to be an early event, arising after a prior oncogenic event[47,48]. In AM, both diploid and WGD tumors were equally likely to have complex rearrangements, indicating that, as has been postulated in sarcomas[24], complex events may be present before WGD in many cases, or occur independently of WGD. In support of this, chromothripsis has recently been shown to occur early in the evolution of AM[49].

We identified a recurrent 4 base pair deletion in *TYRP1* that is predicted to cause a premature termination 31 amino acids downstream. *TYRP1* is a target gene for microphthalmia-associated transcription factor[50] and proposed to play a role in the survival response to oxidative stress[51]. *TYRP1* is also postulated to have a non-coding function, regulating gene expression by

acting as a sponge for miRNAs including miR-16[52]. Further work is required to confirm these potential roles of *TYRP1* in cancer development.

WGS allowed a more complete overview of all forms of somatic aberrations, including the effects of rearrangements which dominated the mutational landscape of many AM. We showed that CM molecular subtypes identified by TCGA have a different distribution in AM, with a higher proportion of *NF1* and Triple WT subtype tumors with many aberrations in the form of structural rearrangements and copy number events. We have previously described[13] how inactivating *NF1* mutations may be underestimated in most studies, and here we confirmed that many *NF1* mutations in AM are predicted LoF rearrangements. The most common subtype, Triple WT was typified by *KIT* SNV mutations, but also *SPRED1* aberrations and amplifications of oncogenes. We identified predicted LoF rearrangements in *SPRED1* that were also associated with poorer survival. Rearrangement breakpoints were also identified upstream of *TERT*, often with nearby super-enhancers and in tumors with higher expression of *TERT* indicating that transcriptional regulation, in addition to copy number, potentially affects telomerase activity in AM. In summary we have shown the potential of WGS for subclassification and identification of therapeutic opportunities in acral melanoma, an uncommon but globally prevalent cancer associated with poor survival.

## Methods

**Human melanoma specimens.** Fresh-frozen tissue and matched normal germline (blood) samples were obtained from the biospecimen bank of Melanoma Institute Australia (MIA) ($n = 83$), QIMR Berghofer Medical Research Institute ($n = 1$), University of Colorado ($n = 3$) and University of Zurich ($n = 1$). All samples were accrued prospectively with written informed patient consent. The study protocol was approved by the Sydney Local Health District Ethics Committee (Protocol No X15-0454 (prev X11-0289) & HREC/11/RPAH/444 and Protocol No X17-0312 (prev X11-0023) & HREC/11/RPAH/32) and cases were approved by institutional ethics committees of Melanoma Institute of Australia, QIMR Berghofer Medical Research Institute (HREC approval P452 & P2274), University of Colorado and University of Zurich. Other information about sample collection and pathology review has been previously described[13]. Thirty-one samples had been previously published and were re-analyzed for this study[13].

**DNA and RNA extraction.** Fresh-frozen tumor DNA was extracted using the AllPrep® DNA/RNA/miRNA Universal kit (Qiagen #80224) for both DNA and RNA extraction. Blood DNA was extracted from whole blood using the QIAamp® DNA Blood Kit (#51126). DNA samples were quantified using a NanoDrop (ND1000, Thermoscientific) and Qubit® dsDNA HS Assay (Q32851, Life Technologies) with DNA size and quality tested using gel electrophoresis. RNA samples were quantified using the Qubit® RNA HS Assay (Q32852, Life Technologies). Most DNA and RNA samples were from the same vial of tissue. Six RNA-seq (MELA_0267, MELA_0015, MELA_0004, MELA_0268, MELA_0068, MELA_0010) samples were the same lesion, but from a different vial of tissue than the DNA sample.

**Whole-genome sequencing.** Sequencing library construction was performed using TruSeq DNA Sample Preparation kits (Illumina, San Diego, California, USA) according to the manufacturer's instructions.

Whole-genome paired-end sequencing was performed on HiSeq2000, HiSeq X Ten or NovaSeq instruments (Illumina, San Diego, CA, USA) at the Kinghorn Cancer Centre, Garvan Institute of Medical Research (Sydney, Australia) or Macrogen (Geumcheon-gu, Seoul, South Korea). Tumor samples underwent whole genome sequencing to a median coverage of 59X (range 35–118) and normal germline samples to a median coverage of 35X (range 21–124). Sequenced data was adapter trimmed using Cutadapt[53] (version 1.9) and aligned to the GRCh37 assembly using BWA-MEM[54] (version 0.7.12) and SAMtools[55] (version 1.1). Duplicate reads were marked with Picard MarkDuplicates (https://broadinstitute.github.io/picard, version 1.129). Tumor purity was assessed using ascatNGS[56] and all tumors had a minimum purity of 35%.

**RNA sequencing analysis.** Libraries were prepared from RNA using the TruSeq Stranded mRNA kit and sequenced with 100 bp paired end reads. RNA-seq reads were aligned using STAR (version 2.5.2a)[57] to the GRCh37 assembly with the gene, transcript, and exon features of Ensembl (release 70) gene model after trimming for adapter sequences using Cutadapt (version 1.9). Quality control metrics were

computed using RNA-SeQC (version 1.1.8)[58] and gene expression was estimated using RSEM (version 1.2.30)[59]. Samples were TMM (trimmed mean of M values) normalized using the R package edgeR[60] and for expression comparisons of samples with and without gene mutations of interest and for PD-L1 (CD274) expression, log2(TMM normalized counts +1) were used. Immune cell deconvolution of the tumor micro-environment was estimated using CIBERSORT[61]. The algorithm was run for 500 permutations using TPM values from RSEM as input with the supplied LM22 (22 immune genes) gene signature file and, as recommended, quantile normalization was disabled.

**Somatic substitution and indel calling**. Somatic SNV and indels were detected using an established pipeline[13]. A dual calling strategy was used to detect SNVs/DNVs/TNVs, with the consensus of two different tools being used for downstream analysis: qSNP (version 2.0)[62] and GATK HaplotypeCaller (version 3.3-0)[63]. Detection of indels (1–50 bp) was carried out using GATK. SnpEff[64] was used to perform variant annotation for gene consequence. Kataegis regions of localized hypermutation were determined using previously established metrics[13]. Inter-mutational distances (the number of base pairs between mutations) were segmented using piecewise constant fitting and putative regions of kataegis were defined as those segments that contained six or more consecutive mutations with a mean inter-mutation distance of ≤1000 bp.

**Determination of sample genetic ancestry**. Genetic ancestry was determined by comparison of acral sample genotypes with the genotypes of populations that were examined in the 1000 genomes project[65]. Analysis was performed using plink version 1.90b6.8. After removal of variants with missing rate of greater than 0.1 or minor allele frequency of less than 0.05, principal component analysis was performed and principal components 1 and 2 were plotted. Samples that did not cluster with a distinct 1000 genome population were classified as Other[19].

**Mutational signatures**. The non-negative matrix factorization (NMF) method described by Alexandrov et al. was used to detect mutational signatures in WGS samples[20]. SigProfiler[20] (using Matlab, version 2016a) was used for the de novo discovery of single base substitution (SBS), dinucleotide (DBS) and indel signatures from the cohort of acral samples. De novo signatures were compared against COSMIC version 3 signatures using cosine similarity. For SBS signatures, the contribution of mutational signatures to individual samples was determined by SigProfilerSingleSample[20], using the signatures identified by de novo analysis as input. For DNVs there were insufficient mutations to extract de novo signatures with confidence. DBS signatures were therefore assigned using deconstructSigs, with a minimum of 15% contribution of mutations required for the signature to be assigned and a limit of 4 signatures per sample. Only samples with greater than 50 mutations underwent assignment, the remaining samples were defined as unassigned/other. For indel signatures, the contribution of each signature identified by de novo analysis was assigned using deconstructSigs, with a minimum of 15% contribution of mutations required for the signature to be assigned. Mutations that could not be assigned by deconstructSigs or signatures that were less than 15% were designated as unassigned/other[66].

**HLA typing and neoantigen prediction**. Class I HLA genotypes were computed for paired tumor-control whole genome datasets using Polysolver (v1.0)[67] and Optitype (v1.3.1)[67] run with default parameters. To avoid discordant calls, predicted HLA-A, HLA-B and HLA-C genotypes through Polysolver were compared with Optitype predictions.

The pVAC-Seq (v4.0.10) pipeline[68] was run using default parameters to predict neoantigens and the NetMHCpan (v4.0) algorithm[69] was used to estimate binding affinity. As recommended, variants were annotated for wild type and mutant peptide sequences with variant effect predictor (v86) (VEP)[70] from Ensembl. Epitopes with binding affinity Inhibitory Concentration ($IC_{50}$) ≤ 500 nM were considered to be potential neoantigens that bind to HLA alleles, and epitopes with $IC_{50}$ of <50 nm had strong binding affinity. The qbasepileup tool (https://github.com/AdamaJava/adamajava/tree/master/qbasepileup) was applied in order to prioritize and identify expressed neoantigens with an $IC_{50} \leq 500$ nM. For each sample, qbasepileup was run in SNP mode to count the reference genome base at each SNP position as well as the mutant bases in the aligned RNA-seq BAM. Duplicates and poorly mapped reads were excluded.

**Significantly mutated gene analysis**. A consensus approach with multiple tools was used to identified significantly mutated genes with respect to substitution and indel mutations. The tools used were OncodriveFML[71], 2020+[72], dNdScv[73] and MutSigCV[74] and all tools were run with default parameters unless otherwise stated. OncodriveFML was executed using CADD v1.0 through the web interface at: http://bbglab.irbbarcelona.org/oncodrivefml/home. For OncodriveFML, MutSigCV and dNdScv, genes were considered significant at a q-value of <0.1. The tool 20/20 + was run for 10000 iterations, using the pre-trained classifier '2020plus_10k.Rdata'. A gene was significant if the oncogene, tumor suppressor gene or driver genes q-values were less than 0.1. A gene was considered to be significantly mutated in AM if it was significant in two or more tools.

**Copy number aberrations and structural rearrangement variants**. Copy number aberrations were identified using ascatNGS[56]. Amplifications were defined as those with a copy number ≥6 and homozygous deletions (copy number 0) were also considered in analyses. Gene-specific copy number was determined by annotation against Ensembl known genes (version 75). Significantly mutated copy number regions were assessed using GISTIC2.0 with a confidence level of 0.95 and a q-value of <0.05. Tumors with whole genome duplication were defined as those with greater than 50% of their autosomal genome having a major copy number (ie the most frequent allele for the segment) that was greater than or equal to two[48]. Chromosomal arm level events of gain or loss were determined by assigning each segment as amplified, deleted or neutral based on whether the total copy number was greater than, smaller than or equal to the ploidy of the sample. A chromosome arm was considered gained or lost if >80% of the total segment lengths for the arm when added together was altered in the same direction[25]. An aneuploidy score was calculated as the sum of the number of autosomal chromosome arms gained or lost together (excluding short arms for the acrocentric chromosomes: 13,14,15,21,22), for a maximum total of 39.

Structural rearrangements were identified using qSV[13]. The potential consequence of the rearrangements, including predicted in-frame gene fusions and LoF variants, was determined using in-house scripts by annotation against Ensembl known genes (version 75). RETREAD[24] (https://github.com/UCL-Research-Department-of-Pathology/RETREAD) was used to identify regions of recurrent breakpoints in 1 Mb bins across the genome. A q-value of <0.2 was considered significant. Localized rearrangements on a per-chromosome basis were identified using previously established metrics[19]: the presence of chromosomes that had a highly significant non-random distribution of breakpoints with a threshold of $p < 10^{-5}$ were considered to be clustered and chromosomes with high numbers of rearrangement events were identified as outliers defined as a breakpoint per megabase rate exceeding 1.5 times the length of the inter-quartile range from the seventy-fifth percentile for each sample with a minimum threshold of 35 breakpoints per chromosome. Chromosomes with at least 10 translocations were considered to have a high number of translocations. Chromosomes that passed one or more of these parameters then underwent a manual review by two people (FN, KN). Chromosomes were defined as having evidence of BFB if there was a clustered region that resulted in loss of telomeres and had a high number of inversions and/or translocations; BFB/chromothripsis if there was evidence for BFB, but also clustered breakpoints, oscillation of copy number and retention of heterozygosity. Regions that did not fit the criteria for BFB or chromothripsis were defined as localized complex. For breakpoints pairs where one breakpoint was within 20 kb upstream of TERT, the proximity of the other breakpoint partner to a super-enhancer (SE) was determined. The complete set of super-enhancers (SE) were downloaded from the dbSUPER enhancer database (accessed 25 July 2019)[38] and super enhancers for melanoma cell lines CJM, COLO679, LoxImVI, SK-MEL-2 and SK-MEL-30 were downloaded from the SEdb database[39] (accessed 11 December 2019) as no acral melanoma-specific data was available. Proximity of a breakpoint to a super-enhancer was defined as being within 100 kb of a SE.

**Rearrangement and copy number signatures**. We used the same statistical framework using NMF that was used for mutational signature analysis for the identification of rearrangement signatures[75]. Rearrangements were classified into the categories used for breast cancer as described by Nik-Zainal and co-workers[23] in the same manner as we have previously applied to a mucosal melanoma cohort[19]. Rearrangements were classified into types of events: deletions, duplications, inversions and inter-chromosomal translocations and further characterized into 32 categories based on size and whether the breakpoints were clustered or non-clustered. Clustered rearrangement breakpoints were defined using the BEDTools cluster function. The de novo signatures identified were compared to the breast cancer signatures using cosine similarity. The R package deconstructSigs was used to estimate the exposure of each identified signature in each sample to reduce overfitting and a minimum of 15% contribution of mutations was required for the signature to be assigned.

CNV signatures were defined using the parameters and methods applied to sarcomas as outlined by Steele and co-workers[24]. CNV signatures were identified from ascatNgs copy number profiles using NMF and 40 categories of events were defined. Segments were classified as heterozygous, LOH or homozygous deletions with further classification by total copy number (0–1 = deleted, 2 = neutral, 3–4 = duplicated, >4 = amplified) and size of the segment (0–0.01 Mb, 0.01–0.1 Mb, 0.1–1 Mb, 1–10 Mb, >10 Mb). NMF was performed using the R package 'NMF'[76] and was run with ranks 2 to 12. To avoid overfitting, NMF was also run with a randomized version of the data for 1000 runs with ranks 2 to 12. Five signatures were chosen as the appropriate rank, based on cosine similarity with the sarcoma signatures and a rank that maximized the consensus silhouette width and the cophenetic distance. In order to reduce overfitting of the exposures to the data, deconstructSigs was used to estimate the exposure of each identified signature in each sample with a minimum of 15% contribution of mutations required for the signature to be assigned. For Fig. 3a, unsupervised clustering was performed using Pearson's correlation coefficient (1−r) as the distance metric and the average clustering method using the proportion (0–1) of rearrangement signatures (RS2, RS4, RS6) and proportions (0–1) of copy number signatures (CNS1, CNS3, CNS5, CNS6, CNS7).

**Telomere length**. Telomere length was estimated from whole-genome sequencing using qMotif[13,19]. Reads with telomeric repeats were counted in both the tumor and matched normal sample and normalized to the mean genomic coverage of the sample. A relative telomere length was expressed as the log2 ratio of read counts in the tumor BAM file to the matched normal BAM file read counts.

**Statistical analysis**. All statistical analyses were performed using R (version 3.5.3) and were two sided, with a p-value or an FDR adjusted p-value of less than 0.05 considered significant. Continuous variables were evaluated between two conditions using Mann–Whitney U-tests. Continuous variables with three or more conditions were calculated using Kruskal–Wallis tests with pairwise Mann–Whitney U-tests with adjustment for FDR to compare between each condition pair[77]. Fisher's exact tests were used to compare categorical variables. The box boundaries of box plots show the first to third quartiles, the median is the center line and the whiskers represent 1.5 times the inter-quartile range. Kaplan–Meier survival curves of melanoma-specific mortality (where people who were alive or who had a death from unknown causes were censored) were compared by log-rank tests. A multi-variate Cox regression model was used to predict melanoma-specific mortality with SPRED1 aberrations, age, sex, overall stage and specimen type included in the model. A multi-variate Cox regression model was used to predict melanoma-specific mortality with the presence or absence of complex chromosomes, age, sex, overall stage and specimen type included in the model. A multi-variate Cox regression model was used to predict melanoma-specific mortality in primary tumors with PTEN mutations, age, sex and overall stage included in the model.

**Reporting summary**. Further information on research design is available in the Nature Research Reporting Summary linked to this article.

## Data availability

Whole-genome sequencing and RNAseq data that support the findings of this study have been deposited in the European Genome-phenome Archive (EGA) and are available under study accession EGAS00001001552 and dataset accession EGAD00001005500. Access to the data can be gained through application to the Data Access Committee for the dataset. Information on how to apply for access is available at the EGA dataset link: https://ega-archive.org/datasets/EGAD00001005500. Data for 1000 genomes in plink2 format are available at: https://www.cog-genomics.org/plink/2.0/resources#1kg_phase. Databases of super-enhancers are available at https://asntech.org/dbsuper/ (DBSuper database) and http://www.licpathway.net/sedb/ (SEdb database). All other data are available in the article, Supplementary Information or available from the authors upon reasonable request.

## Code availability

In-house tools that were used in this publication are available from the Github public code repository under the AdamaJava project (https://github.com/AdamaJava/adamajava).

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

## Acknowledgements

This work was supported by a National Health and Medical Research Council of Australia (NHMRC) Program Grant (1093017, G.J.M., R.A.S., N.H., G.V.L., J.F.T.), an NHMRC project grant (APP1123217) and NHMRC Fellowship grants (R.A.S., N.K.H. - APP1139071, G.V.L.). G.V.L is supported by an NHMRC Practitioner Fellowship and the University of Sydney Medical Foundation. R.A.S is supported by an NHMRC Practitioner Fellowship. J.S.W. is supported by a NHMRC early career fellowship (1111678). N.W. is supported by an NHMRC Senior Research Fellowship (1139071). N.K.H. is supported by an NHMRC Senior Principal Research Fellowship (1117663). P.M.F. was supported by the Deborah and John McMurtrie MIA Pathology Fellowship. T.J.D. was supported by the Jani Haenke Melanoma Pathology Fellowship. Support from Melanoma Institute Australia, the Royal Prince Alfred Hospital and New South Wales Health Pathology is also gratefully acknowledged.

## Author contributions

F.N. was responsible for writing the manuscript. F.N., K.N. were responsible for data interpretation. N.K.H., G.J.M, R.A.S. conceptualized the study. N.K.H, N.W., J.V.P obtained funding for the study. K.N., P.A.J, L.T.K, V.A., P.M. Q.X., A.M.P., V.L., S.H.K., S.W. C.L., O.H. performed data processing and analysis. N.B., C.M.A. were responsible for sample processing and extraction. N.K.H., J.V.P., N.W., W.A.R. acted in a supervisory capacity. N.B., W.A.R., M.P.L, G.V.L., R.V.R., J.F.T.,T.J.D, P.M.F., J.R.S., A.J.S., R.P.M.S., J.S.W, C.M.A, T.M., R.V.G., K.L. and R.G. were responsible for patient data curation, recruitment and tumor acquisition. All authors reviewed and edited the manuscript.

## Competing interests

J.V.P. and N.W. are founders and shareholders of genomiQa Pty Ltd, and members of its Board. R.A.S. has received fees for professional services from Merck Sharp & Dohme, GlaxoSmithKline Australia, Bristol-Myers Squibb, Dermpedia, Novartis Pharmaceuticals Australia Pty Ltd, Myriad, NeraCare and Amgen. G.V.L is consultant advisor for Aduro, Amgen, Array, BMS, MERCK MSD, Novartis, Pierre-Fabre, Roche, Sandoz. R.P.M.S has participated in advisory boards for MSD, Novartis and received speaking honoraria from BMS. J.F.T. has received honoraria for advisory board participation from Merck Sharpe Dohme Australia and Bristol Myers Squibb Australia. J.F.T. has also received honoraria and travel expenses from GSK and Provectus Inc. The remaining authors declare no competing interests.
