## [Peer Review File · Nature Communications]

REVIEWER COMMENTS

Reviewer #1 (Remarks to the Author):

This manuscript reports findings from WGS and transcriptomic profiling of melanomas from the hands and feet. To provide context, prior excellent work by Hayward et al. in Nature (Hayward, N. K., et al. (2017). "Whole-genome landscapes of major melanoma subtypes." Nature 500: 415.) delineated the mutational landscape of cutaneous (CM) and acral melanoma (AM). In that paper, WGS of AM (n=31) was performed, and key findings in AM included: 1) The presence of structural rearrangements including complex rearrangements 2) frequent copy number variation including amplified regions and aneuploid genomes 3) such amplification affecting genes including CCND1, and 4) far fewer mutations per megabase compared to CM, with a relative absence of a UV signature.

In this paper, an additional 56 melanomas and one cell line have been added, and analysed with the original 31 prior published datasets.

Strengths

- 1) Whole genome profiling of melanomas of the nail unit, palms and soles have now been done in additional samples, some coupled with transcriptomic profiling. These additional samples offer the potential for novel driver gene discovery.
- 2) Systematic approach to analysis of WGS data, including mutational signature analysis, copy number signature analysis and rearrangement signature analysis. Whole genome duplication, aneuploidy and complex rearrangement findings are validated for AM.
- 3) Mutations in TYRP1 are of interest in a small (n=7) subset of AM.

Weaknesses

Unlike the prior paper, this work does not however appear to offer a conceptual advance into the understanding of the biology of AM. Specific major points that need to be considered are:

1) The categorisation of AM as a group, whilst conventional, does not take into account the distinct biological contexts of the nail unit vs. palm and sole sites. When not grouped together, the nail unit MMs are n=21, and sole of foot and palm of hands n= 66. Would it not be important to study these categories separately and report these analyses? This is also suggested by the finding that the genomic changes in the nail unit MMs are distinct

2) A further issue is the grouping of primary, recurrent and metastatic tumours. All 3 may be biologically distinct, and each state may carry specific mutational drivers. There are only 36 primary tumors in this study, with 24-27 on sole of the foot, 3 on the palmar surface of the hand and 5 on the nail unit. It may be due to small numbers that these primary tumours were not analysed as a group separate from metastatic and recurrent tumours, but this categorisation may offer important insights.

3) There is no patient level identifier in Supplementary Table S1, making it hard to know if every patient in this study provided only 1 sample (I suspect this is the case based on the control sample IDs?). Also, grouping of recurrent (which may be local at the scar site) and metastatic melanomas (which maybe hepatic) weakens the interpretation – can this information be provided – location of metastasis – eg nodal, bone etc?.

4) Lack of clinical detail correlating with genomic findings. The authors provide summary level data which is confirmatory between melanoma stage and survival in supplementary figure 1. There is no correlation between clinical data and genomic findings reported, with the exception of SPRED1 mutation in 8 tumors that had mutations co-occurring in NF1/NRAS or KIT. Due to small numbers, it

is hard to determine how much of this is attributable to SPRED1, which on its own has a non-significant increase statistically.

Minor:

1) Melanoma of the back of the hand, and the top (dorsum) of the foot would be expected to be similar to cutaneous melanoma (CM) arising on the forearm and shin respectively. 11 of the foot tumours may be CM by the data provided in ST1, and this should be clarified to be from the sole. For example on page 4 line 92, how many of the 14 tumours with high TMB overlap with the 11 on the feet? (Note in the figures and text, "foot" is used when it should explicitly state sole of foot, and "hand" when it should state palm of hand).

2) Are the control DNA samples from blood/saliva, or are they from perilesional normal tissue? The latter may include somatic mosaic mutations which may not be lost following the subtractive mutation calling. The source of the control DNA (eg blood, buccal swab) should be indicated in Supp Table 1.

Reviewer #2 (Remarks to the Author):

Nature Communications Acral Melanoma Whole Genome Sequencing

In this manuscript, the authors present analysis of whole genome sequencing data for 88 acral melanomas and matched normal tissue, predominantly from patients with European ancestry. Acral melanoma is a genetically unique subtype of melanoma and the current study confirms and builds upon what was previously known about the underlying genomic alterations in acral melanoma. The authors perform rigorous analysis of the data, present their findings with clear illustrations and limit their claims to those supported by the data. They identify UV signature in a subset of melanoma, confirm unique features of BRAF V600E acral melanoma, identify a novel recurrently mutated gene, and classify tumors of SNV and SV signatures. Their discussion touches on interesting aspects of tumor genetics and the manuscript is well-referenced.

I think this paper is worthy of publication in Nature Communications.

Minor Comments:

1. Would be nice to mention sequencing depth statistics in the first or second paragraph of the results, as SNV and SV detection sensitivity and specificity may be impacted by depth of sequencing.

2. Reference 15 should be cited in line 163.

3. Ablain et al Science 2018 should be cited regarding SPRED1 in mucosal melanoma.

4. The data presented in 5a and 7a is overlapping and these figures should probably be merged or one put in the supplemental data.

Response to referees

Reviewers' questions and comments are in black, bold text and the authors' responses are in blue text. In the manuscript and supplementary information files, changes are shown with yellow highlighting. In order to accommodate the new analyses and meet the word count requirements for Nature Communications, some text in the previous version of the manuscript has been deleted (shown in track changes).

Reviewer #1:

As we were addressing the questions from reviewer one, additional information came to light about 2 tumour samples. One patient, (MELA_0312), had a primary melanoma involving the ipsilateral ankle region prior to developing their acral melanoma. The sequenced tumour was an ipsilateral groin metastasis and it is uncertain whether this metastasis originated from the ankle cutaneous melanoma or the acral melanoma. Given the sequencing data (high mutation burden and high contribution of UVR signature), the former appears more likely. In light of this additional information, it was decided that this sample should be removed from the analysis, leaving 87 WGS samples (63 with RNAseq). Additional information was also obtained about another tumour previously assigned a site of the toe (MELA_0295) and this tumour was able to be reclassified as being of subungual origin (toenail), giving an updated number of 22 subungual samples. All analyses were redone in light of this new information and the text, figures, supplementary tables and p-values have been updated. There were no major changes to the overall results, with the following exceptions:

- Overall stage displayed only a trend for association with melanoma specific mortality ($p=0.06$), so this figure - previously Supplementary Figure 1b - was removed.
- KIT was previously identified as a SMG, with the new analysis, it was no longer significant (likely due to the fact the removed MELA_0312 tumour had a KIT mutation). NOTCH2 is now identified as a SMG. The text and figures have been updated to reflect the new information about these genes.

This manuscript reports findings from WGS and transcriptomic profiling of melanomas from the hands and feet. To provide context, prior excellent work by Hayward et al. in Nature (Hayward, N. K., et al. (2017). "Whole-genome landscapes of major melanoma subtypes." Nature 500: 415.) delineated the mutational landscape of cutaneous (CM) and acral melanoma (AM). In that paper, WGS of AM ($n=31$) was performed, and key findings in AM included: 1) The presence of structural rearrangements including complex rearrangements 2) frequent copy number variation including amplified regions and aneuploid genomes 3) such amplification affecting genes including CCND1, and 4) far fewer mutations per megabase compared to CM, with a relative absence of a UV signature.

In this paper, an additional 56 melanomas and one cell line have been added, and analysed with the original 31 prior published datasets.

Strengths

- 1) Whole genome profiling of melanomas of the nail unit, palms and soles have now been done in additional samples, some coupled with transcriptomic profiling. These additional samples offer the potential for novel driver gene discovery.
- 2) Systematic approach to analysis of WGS data, including mutational signature analysis, copy number signature analysis and rearrangement signature analysis. Whole genome duplication, aneuploidy and complex rearrangement findings are validated for AM.
- 3) Mutations in TYRP1 are of interest in a small (n=7) subset of AM.

Weaknesses

Unlike the prior paper, this work does not however appear to offer a conceptual advance into the understanding of the biology of AM. Specific major points that need to be considered are:

- 1) The categorisation of AM as a group, whilst conventional, does not take into account the distinct biological contexts of the nail unit vs. palm and sole sites. When not grouped together, the nail unit MMs are n=21, and sole of foot and palm of hands n= 66. Would it not be important to study these categories separately and report these analyses? This is also suggested by the finding that the genomic changes in the nail unit MMs are distinct.

We have carried out additional analyses of subungual and foot/hand subgroups. We have performed the following additional separate analyses for subgroups of subungual or foot(sole)/hand(palm) tumours and amended the text:

- Significantly mutated gene analysis of only subungual or foot/hand tumours subgroups was performed. No new significantly mutated genes were identified. The genes that were significant when analyzing subungual or foot/hand subgroups are listed in Supplementary Table 4.
- Analysis of recurrent copy number events by GISTIC (Supplementary Figure 8a,b). Broadly similar patterns were observed when analyzing subungual or non-subungual foot/hand subgroups. Subungual tumours were significantly amplified in the regions 15q26.3 and the region on chromosome 4 including KIT, whereas non-subungual foot/hand tumours were not. Of 7 tumours with KIT amplifications, 4 were subungual.
- RETREAD analysis to identify recurrent regions of rearrangement breakpoints in subungual or foot/hand only tumours (Supplementary Figure 9a,b). No significant regions were identified in subungual tumours, likely due to the small numbers analysed (n=22), but the pattern of rearrangements was broadly similar when compared with non-subungual tumours from the foot and hand. SPRED1 rearrangements were more common in subungual tumours (32%) than in tumours from non-subungual sites (11%, Fisher's exact test, p=0.039).
- Additional associations were also reported in the text (added text is in italics):
 - "While BRAF V600E mutated tumours were of lower tumour thickness and without ulceration at the time of diagnosis, most BRAF V600

positive tumour samples were recurrence/metastasis specimens (14/16) and were not from subungual sites (1/16 was subungual).”

- *CDK4 aberrations were more common in subungual tumours (Fisher’s exact test $p=0.017$, 32% of subungual, 9% of tumours from other acral sites) with 6/7 subungual CDK4 aberrations being from the toenail.*
- Additional text added to the discussion:
 - Although not confined to a single molecular subtype, subungual tumours lacked BRAF and PTEN mutations and were more likely to have SPRED1 rearrangements or CDK4 aberrations.

2) A further issue is the grouping of primary, recurrent and metastatic tumours. All 3 may be biologically distinct, and each state may carry specific mutational drivers. There are only 36 primary tumours in this study, with 24-27 on sole of the foot, 3 on the palmar surface of the hand and 5 on the nail unit. It may be due to small numbers that these primary tumours were not analysed as a group separate from metastatic and recurrent tumours, but this categorisation may offer important insights.

We have performed additional analyses of primary and recurrence/metastasis tumours as subgroups. As only 3 tumours were local recurrence (and were considered clinically to represent local metastases), these and metastases were combined for analysis.

- Significantly mutated gene analysis of only primary or recurrence/metastasis tumours subgroups was performed. No new significantly mutated genes were identified. The genes that were significant when analyzing primary or recurrence/metastasis separately are listed in Supplementary Table 4 and in the manuscript text.
- Analysis of recurrent copy number events by GISTIC (Supplementary Figure 8c,d). Similar focal amplification and deletion regions were observed between primary tumour and recurrence/metastasis subgroups. However, focal amplifications on chromosome 22p (including EP300) were only significant in primary tumours, with EP300 amplifications occurring more often in primary tumours (31%) than recurrence/metastasis tumours (8%) (Fisher’s exact test, $p=0.0087$).
- RETREAD analysis to identify recurrent regions of SV breakpoints of primary only and recurrence/metastasis only tumours (Supplementary Figure 9c,d). When comparing primary and recurrence/metastasis tumours subgroups, the overall distribution of rearrangement breakpoints was mostly consistent. A new recurrent region of rearrangement breakpoints was identified in primary tumours on 4q34.3. This region contains a long non-coding RNA, LINC00290, which has been reported to be a recurrent deletion site in a pan-cancer study¹⁸ and has been suggested as a common fragile site³⁵. Breakpoints in RBFOX1 (16p13.3) were also more common in primary tumours (50%) than recurrence/metastasis tumours (20%, Fisher’s exact test, $p=0.005$).
- An additional association was added to the text:
 - SPRED1 aberrations were more common in primary tumours (Fisher’s exact test, $p=0.022$, 10% recurrence/metastasis, 31% primary).
- Survival analysis of primary tumours alone (see question 4 below for additional information)
- Additional text added to the discussion:

- The genomic features of primary and recurrence/metastasis tumours were broadly similar, although some aberrations, including EP300 amplifications, SPRED1 aberrations and rearrangements in the regions of RBFOX1 and LINC00290 were more common in primary tumours. Given the small sample size of primary tumours (n=36), a study comparing larger cohorts of primary and recurrence/metastasis tumours would be of interest to further understand any differences.

3) There is no patient level identifier in Supplementary Table S1, making it hard to know if every patient in this study provided only 1 sample (I suspect this is the case based on the control sample IDs?). Also, grouping of recurrent (which may be local at the scar site) and metastatic melanomas (which maybe hepatic) weakens the interpretation – can this information be provided – location of metastasis – eg nodal, bone etc?.

The first column in Supplementary Table 1 was the patient level identifier, however this was unclear due to the way the column was labelled (Sample ID). This has been re-named to “Donor ID” to clarify this. There was only one sample from each donor in the study. The type of the metastasis/recurrence in the relevant samples has been provided in Supplementary Table 1 in the column named “Metastasis/recurrence type” (eg regional lymph node, ITM - in-transit metastasis). Only 3 tumours were a local recurrence; each was associated with histologically clear surgical margins on the primary tumour excision specimen and the recurrence was therefore considered clinically and pathologically to represent a local metastasis. Hence, the local recurrence and metastases samples were combined for analysis.

4) Lack of clinical detail correlating with genomic findings. The authors provide summary level data which is confirmatory between melanoma stage and survival in supplementary figure 1. There is no correlation between clinical data and genomic findings reported, with the exception of SPRED1 mutation in 8 tumours that had mutations co-occurring in NF1/NRAS or KIT. Due to small numbers, it is hard to determine how much of this is attributable to SPRED1, which on its own has a non-significant increase statistically.

We agree that with respect to survival, there was an overall lack of correlation with genomic findings. There were however, associations of genomic findings with other clinical parameters such as tumour thickness and the presence or absence of ulceration, as described in the text. In this cohort, survival analysis is confounded by different treatment regimens in a quarter of the cohort, as well as the different tumour stages and the presence of both primary and recurrence/metastasis tumours. Additional survival analysis was performed, though the power for primary tumours alone was limited due to small sample size (n=36). The following results are now reported in the text:

- Survival analysis using only primary tumours identified PTEN mutated tumours as having poorer survival, however there were only a small number of primary tumours (n=5) that had PTEN mutations (Supplementary Figure 11).

- Survival analysis of whether tumours have evidence of complex events or not, show that tumours with complex events have a longer melanoma-specific survival (Supplementary Figure 4a,b).

Minor:

1) Melanoma of the back of the hand, and the top (dorsum) of the foot would be expected to be similar to cutaneous melanoma (CM) arising on the forearm and shin respectively. 11 of the foot tumours may be CM by the data provided in ST1, and this should be clarified to be from the sole. For example on page 4 line 92, how many of the 14 tumours with high TMB overlap with the 11 on the feet? (Note in the figures **and text, “foot” is used when it should explicitly state sole of foot, and “hand” when it should state palm of hand**).

- With one exception, all acral tumours listed as being from the foot or hand were confirmed by a pathologist to have arisen in acral-type skin and are therefore confirmed to be from the sole of the foot or the palm of the hand. The exception was MELA_0312 noted above; the patient who also had a prior cutaneous melanoma and therefore has now been removed from analysis (see first paragraph of the response to Reviewer #1). Supplementary Table 1 has been updated to specify Foot - sole for the relevant samples.
- A sentence was updated in the first paragraph of the results:
 - Fifty-nine tumours were from the sole of the foot, six were from the palm of the hand and there were twenty-two subungual tumours (15 toenail, 7 thumbnail/fingernail).
- Updated figures to specify Foot (sole) and Hand (palm)
- For the 14 tumours with high TMB, one was originally referred to as being from the foot without further specification (now updated to Foot – sole) and one was the removed tumour MELA_0312. The remaining 12 were from the fingernail (7), sole of foot (3 – one with a mismatch repair signature to explain the high mutation burden), palm of the hand (1) and toenail (1).

2) Are the control DNA samples from blood/saliva, or are they from perilesional normal tissue? The latter may include somatic mosaic mutations which may not be lost following the subtractive mutation calling. The source of the control DNA (e.g. blood, buccal swab) should be indicated in Supp Table 1.

Control DNA samples are from blood. The source of the control DNA has been added to Supplementary Table S1 in the column “Germline specimen type”.

Reviewer #2:

In this manuscript, the authors present analysis of whole genome sequencing data for 88 acral melanomas and matched normal tissue, predominantly from patients with European ancestry. Acral melanoma is a genetically unique subtype of melanoma and the current study confirms and builds upon what was previously known about the underlying genomic alterations in acral melanoma. The authors perform rigorous analysis of the data, present their findings with clear illustrations and limit their claims to those supported by the data. They identify UV signature in a subset of melanoma, confirm unique

features of BRAF V600E acral melanoma, identify a novel recurrently mutated gene, and classify tumors of SNV and SV signatures. Their discussion touches on interesting aspects of tumor genetics and the manuscript is well-referenced.

I think this paper is worthy of publication in Nature Communications.

1. Would be nice to mention sequencing depth statistics in the first or second paragraph of the results, as SNV and SV detection sensitivity and specificity may be impacted by depth of sequencing.

Sequencing depth statistics have been added to the results in the first paragraph.

2. Reference 15 should be cited in line 163.

Reference 15 is now cited as suggested on page 6.

3. Ablain et al Science 2018 should be cited regarding SPRED1 in mucosal melanoma.

Ablain et al Science 2018 has now been added as a reference and is cited on page 10.

4. The data presented in 5a and 7a is overlapping and these figures should probably be merged or one put in the supplemental data.

Figure 5 has been removed as a main figure, and is instead included as Supplementary Figure 7.

REVIEWERS' COMMENTS

Reviewer #1 (Remarks to the Author):

Thank you for performing the additional analyses and clarifying the points raised with additional information. I have no further points to raise.